# Evaluation of Five *Chrysanthemum morifolium* Cultivars against Leaf Blight Disease Caused by *Alternaria alternata* at Rooting and Seedling Growth Stages

**DOI:** 10.3390/plants13020252

**Published:** 2024-01-16

**Authors:** Mayada K. Seliem, Naglaa A. Taha, Nahla I. El-Feky, Khaled Abdelaal, Hassan El-Ramady, Mohammed E. El-Mahrouk, Yousry A. Bayoumi

**Affiliations:** 1Ornamental and Floriculture Department, Horticulture Research Institute, El-Sabahia, Alexandria 21599, Egypt; mayadaseliem@gmail.com; 2Plant Pathology Research Institute, Agriculture Research Center, Giza 12619, Egypt; naglaa_abdelbaset@yahoo.com (N.A.T.); nahlaelfeky2@gmail.com (N.I.E.-F.); 3EPCRS Excellence Center, Plant Pathology and Biotechnology Lab., Faculty of Agriculture, Kafrelsheikh University, Kafr El-Sheikh 33516, Egypt; khaled.elhaies@gmail.com; 4Soil and Water Department, Faculty of Agriculture, Kafrelsheikh University, Kafr El-Sheikh 33516, Egypt; 5Horticulture Department, Faculty of Agriculture, Kafrelsheikh University, Kafr El-Sheikh 33516, Egypt; threemelmahrouk@yahoo.com

**Keywords:** florist’s daisy, phytopathogen, antioxidants, photosynthetic rate, climate change, chlorophyll fluorescence, histological parameters

## Abstract

During the winter of 2018, leaf blight on florist’s daisy (*Chrysanthemum morifolium* L.) was noticed in Egypt. The disease, which was identified as caused by *Alternaria alternata*, was widely spread and led to serious damage for the exportation sector of this crop. Therefore, a study was conducted to better understand what can be conducted to minimize the problem in the future. Isolates were gathered and evaluated on five chrysanthemum cultivars (i.e., ‘Feeling Green Dark’, ‘Talitha’, ‘Chrystal Regan’, ‘Arctic queen’, and ‘Podolsk Purple’) grown in a greenhouse. The objectives were to isolate and identify the phytopathogen and detect the resistant degree of these cultivars with emphasis on the early growth stages of the crop. The results showed that ‘Podolsk Purple’ was the most resistant cultivar against the different isolates during the rooting and seedling growth stages. ‘Chrystal Regan’ was very susceptible to the different isolates. In addition, the isolate from ‘Feeling Green Dark’ was the strongest, which negatively affected the chlorophyll content and its fluorescence parameters besides other measured vegetative and anatomical features. The findings indicated that the best anatomical characters of the stem and leaf, like the thickness of cuticle and cortex, stem diameter, xylem vessel diameter, and thickness of epidermis as well as lamina thickness were recorded in the ‘Podolsk Purple’ cultivar. This study highlighted that by using the right cultivars, chrysanthemum can be cultivated during the winter season under Egyptian conditions. These results can be a part of solution to overcome the leaf blight caused by *A. alternata* on chrysanthemum during the early growing stages.

## 1. Introduction

Chrysanthemum or florist’s daisy (*Chrysanthemum morifolium* Ramat.), Asteraceae family, represents one of the most economically important subtropical ornamental perennial plants worldwide [1]. The plant originated in China and is considered the second most valuable flowering crop after rose [2]. Chrysanthemum products have several uses, with both ornamental and herbal, medicinal applications and the plant can also be used as tea and food [3]. The medicinal and therapeutical values can be linked to bioactive constituents (e.g., flavonoids, terpenoids, volatiles oils, polysaccharides, and steroids) with claimed pharmacological activities such as an anti-obesity effect, anti-inflammatory activity, anticancer activity, cardioprotective effect, neuroprotective effect, and antidiabetic [3,4,5,6]. Due to the high economic value of chrysanthemum, especially in the global trade of cut flowers, the plant growth and development should be protected from different phytopathogens, which impact mainly the leaf growth, such as leaf blight [7], and leaf spot disease caused by *Nigrospora oryzae* [8].

Due to the ongoing changing climate with an increasing atmospheric temperature, many plant diseases can attack *C. morifolium*, and the fungal diseases especially can cause considerable losses in productivity and quality as well as the medicinal value [9]. These diseases may include powdery mildew caused by *Golovinomyces* sp. [10], leaf blight caused by *A. alternata* [11,12] or by *Nigrospora sphaerica* [7], leaf spot caused by *Nigrospora oryzae* [8], root disease caused by *Fusarium oxysporum* [13], the least rust by *Puccinia horiana* [14], and the Verticillium wilt by *Verticillium dahliae* [15]. Many studies have been published about the fungal diseases as epidemic diseases caused by rust and *Verticillium* spp. on chrysanthemum such as Sumitomo et al. [16], Bi et al. [17], Gao et al. [18], Munilakshmi et al. [19], and Chen et al. [20]. It is reported that *V. dahliae* can colonize the vascular tissues of infected plants which may persist in the soil for up to more than 10 years, even in the absence of a host [15]. Chrysanthemum white rust caused by the fungus *Puccinia horiana* has become an endemic disease to most chrysanthemum-growing areas in China [18]. Thus, several pathogenic micro-organisms can infect chrysanthemum leading to a serious decline in the productivity [13].

Among these phytopathogens, the necrotrophic fungus *Alternaria* spp. can be destructive, causing black spot or leaf blight disease [9]. Leaf blight disease has severe impacts on chrysanthemum, which results in invaded *Alternaria alternata* that attacks mature leaves, with a rapid spread through plant tissues, soil, and surrounding air, especially under high temperature conditions [21]. *A. alternata* can cause serious damage after infection by secreting fungus toxins and enzymes for cell wall degrading in chrysanthemum tissues. The damage of this pathogen goes back to the need to obtain nutrients from decaying tissue leading to round spots with dark mildew layers [12]. Consequently, leaf blight disease caused by *A. alternata* is considered one of the most damaging diseases, leading to losses more than 80% in the yield as well as the deteriorative impact in the quality of producing flowers [11]. *A. alternata*, as a common fungus species, is considered pathogenic in several important crop plants or an endophytic for many plants [22]. In many studies and with various plant species, anatomical parameters have been recorded as important to explain the changes in growth under various conditions including, water deficit [23,24], salt stress [25,26], and plant pathogens [27,28,29]. These changes in anatomical characters may help the plant to cope with stressful conditions and better tolerate plant pathogens.

Leaf blight disease on chrysanthemum has been reported in many countries around the world, such as New Zealand [30], India [31], Mexico [32], South Africa [33], United States [34], and Egypt [35]. Therefore, numerous works have been conducted to display the variations among various cultivars of chrysanthemum in the response of leaf blight (in the term of resistant and susceptible), as well as the degrees of infection of these cultivars [11,31,36]. It is also believed that the leaf blight disease was introduced into Egypt through the importation of cuttings taken from mothers of infected chrysanthemum with the disease during the years of 2018 [35]. Egypt contains many commercial cultivars of chrysanthemum, which are produced in greenhouses during the year and exported abroad to Saudi Arabia, Libya, the Emirates, and Kuwait. Consequently, the disease led to a deterioration in production and a decrease in quality, which led to huge losses for farmers. Therefore, there was a need to search for cultivars that could be resistant to this disease and that could be used in production throughout the crop seasons.

The first successful step in chrysanthemum production is producing seedlings free of disease. Therefore, this investigation creates a main impact to the examination on leaf blight disease caused by *A. alternata* on different cultivars of *C. morifolium* by demonstrating different responses towards this disease during both rooting and seedling growth stages. This response was also confirmed by measuring different vegetative parameters, photosynthetic rate, and anatomical features of leaves and stems. There are several important areas where this study makes an original contribution. For farmers, it is important to choose the appropriate cultivars which are tolerant to the current isolates of leaf blight disease under Egyptian conditions. This is important to achieve the highest yield and quality. It was also important to examine whether the different isolated strains of leaf blight disease infected different cultivars to the same degree.

## 2. Materials and Methods

### 2.1. Plant Resources and Experimental Site

Five disease free cultivars of chrysanthemum (namely ‘Feeling Green Dark’, ‘Talitha, Chrystal Regan’, ‘Arctic queen’, and ‘Podolsk Purple’) were obtained from a private nursery (in El-Gharbia Governorate, Egypt) for the current study. In addition, four infected cultivars (i.e., ‘Feeling Green Dark’, ‘Talitha’, ‘Chrystal Regan’, and ‘Arctic queen’) were collected from another private nursery (in Menoufia Governorate, Egypt) for *Alternaria* isolation. All experiments were carried out in the farm of the protected cultivation center, Department of Horticulture, Faculty of Agriculture, University of Kafrelsheikh, Kafr El-Sheikh, Egypt. The experiments were repeated at two different times (on the 6 January and 6 February 2023 for the first and the second time, respectively).

### 2.2. Description of the Chrysanthemum Varieties

Chrysanthemum cultivars used for the experiments are cultivated during the entire crop season in all areas of Egypt for cut or potted flowers production. Talitha is preferred to be planted in the winter during October, whereas Arctic queen is preferable in the summer during April. The cultivars used in this study differ in terms of plant height, stem diameter, and flower color. Concerning Feeling Green Dark, it has a high, strong stem with large thickness and green flowers, whereas Talitha is characterized by a medium-height stem with thin thickness and single, light purple flowers. Regrading Chrystal Regan, Arctic queen, and Podolsk Purple, they are characterized by strong stems with a large thickness and their flowers are in light red colors, white, and purple, respectively. The percentage of cultivation of studied cultivars varies within different farms according to the planting season and the marketing contract. In general, the cultivation rates of Feeling Green Dark, Talitha, Chrystal Regan, Arctic queen, and Podolsk Purple cultivars were 3–6, 3–5, 8–10, 20–25, and 10–15%, respectively, as an average for 10 farms in the Menoufia and Gharbia governorates.

### 2.3. Isolation, Purification, and Identification of the Pathogens

The four isolates as diseased fungi were collected and isolated from diseased plants with observed symptoms, with *Alternaria* leaf spot indicators. Infected leaves were collected, washed using tap water, censored in small slices (5 mm), external sanitized for 2–3 min in a solution of 0.5% sodium hypochlorite, and finally washed thrice with sanitized distilled water. Slices were placed between two layers of sterilized filter papers to dry and cultured on potato dextrose agar (PDA) medium in Petri dishes (9 cm) at 28 ± 2 °C for 72 h. Pure cultures were used for collection for all four pathogenic isolates via the hyphal tip technique. The four isolates were recognized and known as *Alternaria alternata* based on their morphological structures and microscopic description, according to the method of Ozcelik and Ozcelik [37] at the Department of Mycology and Disease Survey, Plant Pathology Research Institute, ARC, Giza, Egypt.

### 2.4. Screening of Chrysanthemum Cultivars against Alternaria alternata Isolates

The cuttings of five chrysanthemum cultivars were planted into Styrofoam trays (209 cell), which were occupied by sterilized combination media of both vermiculite and coco peat (equally by volume) and maintained in a shading greenhouse (50% shading). All cultures were kept under a continuous light condition into photosynthetic photon flux density at 100 µmol m^−2^ s^−1^ during the rooting stage. All studied cultivars were planted in trays using a rate of 30 cuttings for each one tray and three replicates (10 cuttings for each replicate). The experiment was arranged in five Styrofoam trays. One day after planting, the cuttings in each tray were infected with one isolate from different four isolates of *A. alternata*. The whole cutting’s trays of five cultivars were fully sprayed with the conidial suspension of 10-day-old culture from *A. alternata* (1 × 10^7^ conidia ml^−1^ in sanitary water) of the four isolates, which were isolated from the four cultivars. The tray cultured with cuttings and treated with sterilized water free conidial was kept as a control. All treatments were covered separately with a plastic tunnel (50 µ) to prevent infection between the different isolates. Following the procedure of Arunkumar et al. [38], the diseased signs were detected weekly afterward the inoculation, and the percent of disease severity was registered at 7, 14, and 21 days after inoculation (DAI). This inoculation was performed using plastic tunnels under a temperature of 25 ± 2 °C, humidity 90 ± 5%, and 14:10 h L/D photocycle. The main steps and treatments during the current study are presented as follows (Figure 1):The disease was discovered in many cultivars in different fields, and four infected cultivars were selected.The infected plant samples were identified to isolate the causal agent of disease, and confirmed.The needed cuttings of non-infected cultivars were brought under the study to propagate in a nursery by treating their roots with a rooting hormone (indole butyric acid, IBA).Only one variety (Podolsk Purple) did not appear to be infected in some fields, so it was focused on to see whether it is actually a resistant variety or not.The selected cuttings were cultivated in trays (209 cell) for enhancing the rooting of the cuttings.Then, an artificial infection for the cuttings by *A. alternata* was performed.The severity of the disease infection of the different isolates on the different cultivars was determined.The susceptibility rate in the cultivars under study was evaluated for to the disease based on the infection degree.The anatomy of leaves and stems were performed to determine the resistance/sensitivity to the disease.

### 2.5. Disease Assessments

The leaf blight disease severity was evaluated weekly at 7, 14, and 21 DAI according to the method of Arunkumar et al. [38] using the following scales; (0) no symptoms, (1) limited spots beside the tip cover 10% leaf area, (2) many dusky brown spots encasement up to 20% leaf area, (3) numerous patches with pallid outer zone encasement up to 40% leaf area, (4) cover up to 40% leaf area, and (5) whole desiccation of the leaves, as shown in Figure 2. Percent disease index (PDI) was assessed from the following formula [39]:$$\mathrm{Percent}\;\mathrm{disease}\;\mathrm{index}\;(\mathrm{PDI})=\frac{Sum\;of\;numerical\;rating\;scores}{No.\;of\;leaves\;per\;plant\;observed\times maximum\;rating}\times 100$$

### 2.6. Vegetative and Photosynthetic Parameters

Growth parameters were determined after 21 days from the plant cuttings including the number of roots and leaves per seedling, root length (cm), and then fresh and dry weights of both the root and seedling (g) per seedling. The chlorophyll content (chlorophyll index) when measuring the greenness in the leaf without destroying them using the SPAD-501 portable leaf chlorophyll meter (Minolta, Tokyo, Japan) was measured according to the method described by Yadava [40] at 21 days from the plant cuttings.

Chlorophyll fluorescence characteristics were determined on the leaf surface. Seedlings were kept in a dusky place for 30 min before the measurements. Modulated fluorescence was recorded with a portable chlorophyll fluorometer (OS-30p, Opti-Sciences, Inc., Hudson, NH, USA). The least fluorescence value (F_0_) was measured for 30 min in dusky-adapted leaves using light of <0.1 µmol m^−2^ s^−1^; however, the greater fluorescence value (Fm) was recorded at 3500 µmol m^−2^ s^−1^ photosynthetic photon flux density in the same leaves. The greater changeable fluorescence value (Fv = Fm − F0) and the photochemical efficiency of PSII (Fv/Fm) were registered for dusky-adapted leaves [41]. Three seedlings were randomly chosen and parameters were completed on the leaves with the leaf chamber. Each treatment was concluded with the use of three single leaves as three replications.

### 2.7. Antioxidant Capacity

Antioxidants capacity in the fully expanded young leaves of seedlings was measured using DPPH method according to Binsan et al. (2008) [42], where 0.15 mM of 2,2-diphenyl-1-picryl hydrazyl (DPPH) (ethanol 95%) was mixed with the solution of protein (0.1%) into 5 mM from the PBS buffer solution pH 7.2) at an equal ratio (1:1 by volume). The concoction was assorted and kept in the dusky place for 30 min at 24–26 °C. The reading of the mixed solution was recorded at 517 nm with a spectrophotometer (Libra S80PC, Biochrom, UK). The calibration curve was arranged using trolox in the range of 12.5 to 100 μM.

### 2.8. Histological Examinations

Sections (5 × 5 mm) were taken from the middle part of the leaf including the main midvein, and were taken from the internode in the second season. Specimens were fixed in solution of formaldehyde–acetic acid–alcohol for 48 h, washed gently with disinfected water, dehydrated in sequences of ethanol, cleared in ethanol: xylene (3:1–1:1–1:3% and 100% xylene), and entrenched in paraffin wax (52–54 °C fusion points). The sections were made at 10–15 μm thickness by rotary microtome, double stained with safranin–light green, cleared in clove oil, and mounted in Canada balsam according to Ruzin [43]. A light microscope was used to examine the chosen sections (five sections from each treatment) to study lamina thickness (μm), upper and lower epidermis, palisade and spongy tissue, as well as the main vascular bundle dimensions (length and width in μm) of leaf mesophyll. The anatomical structure of the stem was also investigated.

### 2.9. Statistical Analyses

Each trial was arranged with complete randomization and using a two-factor model. The first factor was the four pathogen isolates and the second factor was the five cultivars. Each trial had three replicates and each experiment was repeated twice. Four Alternaria isolates were arranged within each replicate which consisted of one Styrofoam tray. Data were averaged and statistically analyzed using a two-way ANOVA approach of the CoStat program (Computer Program Analysis, CoHort Software, Version 6.45, Monterey, CA, USA). Duncan’s multiple range tests at 5% probability levels were used compare among the means [44].

## 3. Results

### 3.1. Identification of Different Isolates of A. alternata

Four Alternaria isolates were isolated from four chrysanthemum cultivars and fungal mycelium of the four pathogenic isolates were grown in vitro on PDA medium in Petri dishes (9 cm). The morphological features and microscopic parameters are shown in Figure 3A,B, respectively. The values of the percent disease index (PDI) are provided in Table 1 and shows the result during the rooting stage under Alternaria blight spot pathogen infection. In general, the PDI (%) increased by time (from 7 to 21 days) after inoculation for all studied cultivars except for the ‘Podolsk Purple’ cultivar, which showed no symptoms of any of the four isolates. The Alternaria pathogen isolated from the ‘Chrystal Regan’ cultivar recorded the highest PDI (92.2%) after 21 days from inoculation. Additionally, the ‘Chrystal Regan’ cultivar gave the highest percent of PDI under all pathogen isolates at 7, 14, and 21 days after inoculation, whereas the cultivar ‘Podolsk Purple’ did not show any infection for all the four isolates, and should be referred as a resistant cultivar without any infection from the examined isolates. Each Alternaria pathogen that was isolated from a given cultivar recorded the highest PDI on the same cultivar. This was especially observed 21 days after the infection (70.8, 84.7, 92.2, and 80.5% of the four infected cultivars ‘Feeling Green Dark’, ‘Taliha’, ‘Chrystal Regan’, and ‘Arctic Queen’, respectively). The same result is shown in Figure 4 and Figure 5, which present the response of chrysanthemum cultivars seedlings to infection by the different isolates 21 days after inoculation during the rooting and seedling stage under greenhouse conditions. Again, ‘Podolsk Purple’ did not show any symptoms at any of the dates or after inoculation of the four Alternaria isolates.

### 3.2. Vegetative Growth and Photosynthetic Parameters

Under low tunnel conditions and after 21 days from planting, the vegetative growth traits, chlorophyll fluorescence, and total chlorophyll content were achieved from chrysanthemum plants during both the rooting and seedling stages (Table 2 and Table 3). All vegetative growth traits were the highest for ‘Podolsk Purple’. Among the susceptible cultivars, ‘Feeling Green Dark’ recorded the highest vegetative parameters compared to the other infected cultivars. Regrading chlorophyll content and its fluorescence parameters, the same trends were observed. Both ‘Podolsk Purple’ and ‘Feeling Green Dark’ can be noticed in Table 3. In general, the values of photosynthetic parameters decreased due to infection compared to the resistant cultivar (‘Podolsk Purple’) which produced the highest growth traits values under the infection of all Alternaria isolates. Both the first and third isolates, isolated from ‘Feeling Green Dark’ and ‘Chrystal Regan’ cultivars, respectively, were the strongest isolates showing the lowest values for chlorophyll content and its fluorescence parameters, as well as the other vegetative growth traits. The lowest vegetative growth traits values were resulted from the ‘Chrystal Regan’ cultivar under all Alternaria isolates, and especially with the third isolate that showed the lowest values.

### 3.3. Antioxidant Capacity

The antioxidant capacity of the studied chrysanthemum cultivars that were exposed to the four isolates is presented in Figure 6. For each isolate, there was a significant difference among cultivars, where the ‘Podolsk Purple’ cultivar provided the highest values under different infected cultivars by isolate no. 4. The other four infected cultivars had a similar trend of antioxidant capacity that ranged from 38 to 48 μM TE/10 g. That means the infected cultivars under different isolates had a similar response to the fungus disease.

### 3.4. Histological Examinations

#### 3.4.1. Anatomical Characters of Stems

The anatomical parameters of infected cultivars by *Alternaria alternata* of stems are tabulated in Table 4, and presented in Figure 7. The studied parameters included the thickness of the cuticle, epidermis, and cortex as well as the diameter of the vascular cylinder, stem, and xylem vessel. The results for these parameters were somewhat surprising given the fact that the Podolsk Purple cultivar showed the highest values in thickness of cuticle (3.94 µm), cortex (99.03 µm), stem diameter (490.76 µm), and xylem vessel diameter (17.09 µm), whereas the lowest value for this cultivar was recorded for the thickness of the epidermis (3.976 µm) compared to the other cultivars. On the contrary, the ‘Feeling Green Dark’ cultivar showed the lowest values of the anatomical traits of the stems, especially thickness of the cuticle (1.67 µm), cortex (52.96 µm), stem diameter (365.98 µm), and xylem vessel diameter (13.42 µm), while the highest value for this cultivar was recorded for the thickness of the epidermis (4.90 µm) in most cases compared with the other cultivars.

#### 3.4.2. Anatomical Characters of Leaves

In the case of the stem, almost all the highest values of all histological parameters were linked to the Podolsk Purple cultivar, but for leaves only the following parameters had the highest values: thickness of the epidermis (14.26 µm) and lamina (68.53 µm). The thickness of leaf fractions is a vital factor (Table 5 and Figure 8). The lowest values in leaf thickness differed from one cultivar to another. The lowest thickness of epidermis (7.63 µm), mesophyll tissue (17.81 µm), and lamina (23.84 µm) belonged Chrystal Regan, whereas the thickness of cuticle (3.08 µm) and middle vein (150.8 µm), as well the vascular bundle diameter (69.15 µm), was noticed for Feeling Green Dark cultivar leaves. Interestingly, the resistant cultivar of Podolsk Purple gave the highest histological traits of epidermis thickness (14.26 µm), mesophyll tissue thickness (45.20 µm), and lamina (68.53 µm) compared to other infected cultivars by the *A. alternata* pathogen.

## 4. Discussion

This section seeks to address the following questions: why was this study was carried out and why was this work focused on this crop, and how was the disease evaluated? Based on the result, further questions arise. What are the suggested methods to control this disease? What is the main sustainable strategy to avoid/minimize the damage from this disease? Finally, what are the main recommendations from this study?

Concerning the significance of the present study, it is designated on chrysanthemum, as an important flower and medicinal crop (Figure 9). The flowers of this crop are very diverse, and mainly differ from one variety to another, mainly in their lignified stems, floral morphologies, and high growth in a wide range of habitats such as rural, urban, and farmland [45]. This crop is considered as important in Egypt and many other countries around the world and functions as a cash crop for exportation, which saves hard currency for the country and provides farmers with income. It was noticed that during 2018, the phytopathogen *A. alternata* spread epidemiologically, reaching to a peak in 2020. This catastrophe led to converting Egypt from an exporter to an importer of chrysanthemum with terrible economic damages. This not only lead to a shock in the Egyptian national economy, but also increased the local market prices of these flowers. From this point, several researchers started to search about different strategies against this disease. These approaches included different control tools and to find resistant cultivars. This was the main justification of the current study.

Two years ago, a survey on infected fields was initiated to bring infected plants with leaf blight disease to a laboratory to verify the cause and identify different isolates. The isolation and identification of this disease was completed and followed by an evaluation of the most common cultivars in Egypt. A complete series of studies on this disease from different points of view were carried out, which started with the evaluation of five cultivars against this phytopathogen as published in the current paper. More studies on the chemical control using different pesticides are ongoing, and will be published soon. In the current investigation, the selected five cultivars were evaluated during the rooting and seedling stages with a focus on vegetative, physiological, and histological parameters. The disease assessment besides the isolation, purification, and identification of the pathogens were totally considered and measured. The infected plants did not form any roots.

Results from the current investigation showed that the variety ‘Podolsk Purple’ was found to be resistant to leaf blight disease. This resistance was confirmed through vegetative, physiological, and anatomical parameters. Concerning the disease parameters, this variety did not show any symptoms of the disease, so it is considered as resistant with a percent disease index recorded as 0.0% for all isolates during the rooting stage. The same good result was also observed at the seedling stage, and for photosynthetic parameters as well as the antioxidant capacity. It is obvious that the cultivar ‘Podolsk Purple’ was the most resistant cultivar with the highest values in all studied vegetative growth parameters both at the rooting and seedling stages, which included the number of roots or leaves per seedling, root length, and the fresh and dry weight seedling. The total chlorophyll content, chlorophyll florescence, and antioxidants capacity were also highest for this cultivar, which supported the resistance against the disease.

Previously, only a few studies have been published on leaf blight disease caused by *A. alternata* on chrysanthemum. This includes a study in Prayagraj, India [36], Nanjing, China [12,21], and Greater Cairo, Egypt [35], the latter using seven cultivars (where ‘Dayangju’, ‘Jinba’, and ‘Chrystal Fresh White’ were among them. In the current study, the histological studies in the cross sections in both leaves and stems showed the highest values of cuticle thickness, cortex thickness, stem diameter, xylem vessel diameter, thickness of leaf epidermis, as well as lamina thickness in ‘Podolsk Purple’. The improvements in anatomical structures such as cuticle and cortex thickness and xylem vessel diameter are of importance as defence mechanisms against biotic and abiotic stresses. This has been shown for drought stress [46], water deficit stress [24,47], salt stress [26], and plant disease stresses [27,29]. The increase in cuticle thickness is an adaptational characteristic to tolerate the stressful conditions and protect the internal tissues [47]. In addition, the enhancement of other anatomical features may provide mechanical support to the plants against various stressful conditions [48,49]. This result agrees with the high value of antioxidant capacity in the ‘Podolsk Purple’ cultivar. The improvement in anatomical characters further agrees with the high value of antioxidant capacity in ‘Podolsk Purple’. Tolerance for fungus diseases also could be attributed to the high levels of antioxidant enzymes. These results may explain why this cultivar is more tolerant to *Alternaria* infection compared to the other infected cultivars. A similar pattern has been observed on wheat plants under biotic stress [28,29]. Further studies are needed on the possible biocontrol of this disease in addition to using the resistant cultivar as identified in the current study.

## 5. Conclusions

As a cash and medicinal crop, chrysanthemum is considered one of the most popular and economically important flowering crops worldwide. This plant has several benefits including nutritional, ornamental, and medicinal attributes. The plant originated from China cultivation, which started more than 1600 years ago, and successively the plant was introduced to Europe, Africa, and other continents. This crop is not only very important for the Egyptian economy, but also for many countries such as China, India, Sri Lanka, and Japan. This study was carried out to identify and evaluate five cultivars of chrysanthemum under infection by the fungus of *A. alternata*. The results showed that ‘Podolsk Purple’ was the most resistant and ‘Chrystal Regan’ was the most sensitive cultivar caused by the four *A. alternata* isolates that were collected in Egypt. Vegetative, physiological, and anatomical features confirmed the above result.

## Figures and Tables

**Figure 1 plants-13-00252-f001:**
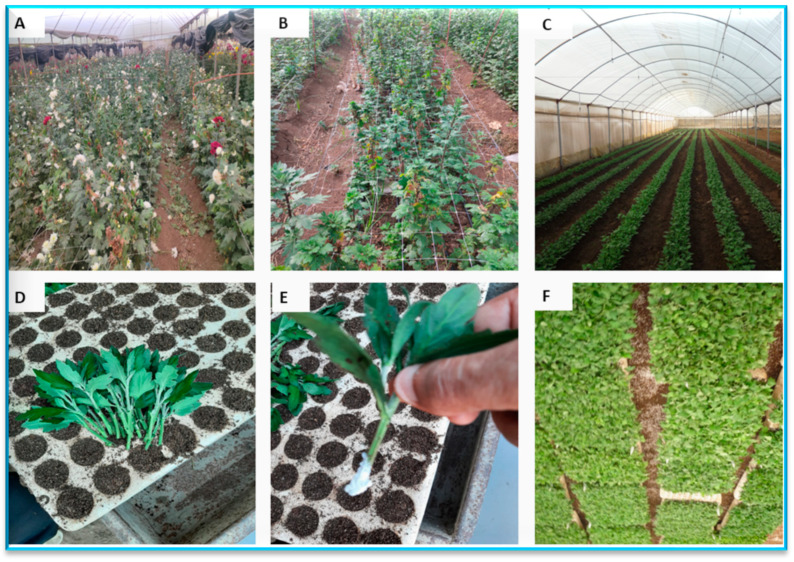
Overall outline of the present investigation with the various experimental stages. (**A**) Infected ‘Arctic Queen’ cultivar, (**B**) infected ‘Chrystal Regan’ cultivar, (**C**) greenhouse of mother chrysanthemum cultivars (non-infected plants), (**D**) production of the cuttings using trays, (**E**) producing the treated cuttings with indole butyric acid (IBA), and (**F**) planted cuttings of the cultivars in trays.

**Figure 2 plants-13-00252-f002:**
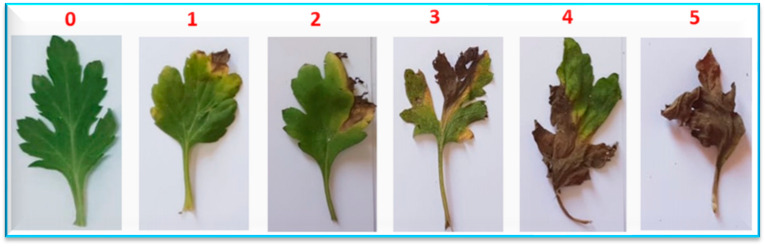
A visual score scale with different classes (from 0 to 5) of leaf blight disease caused by *Alternaria alternata*.

**Figure 3 plants-13-00252-f003:**
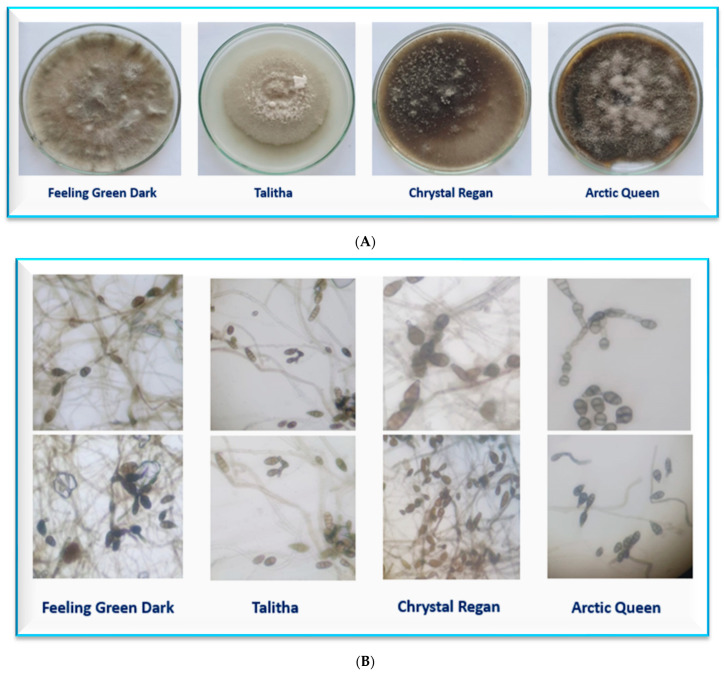
Mycelial growth features variability of different fungal colonies of *A. alternata* isolated from four chrysanthemum cultivars (**A**); the microscopical features variability of studied different fungal isolates (**B**).

**Figure 4 plants-13-00252-f004:**
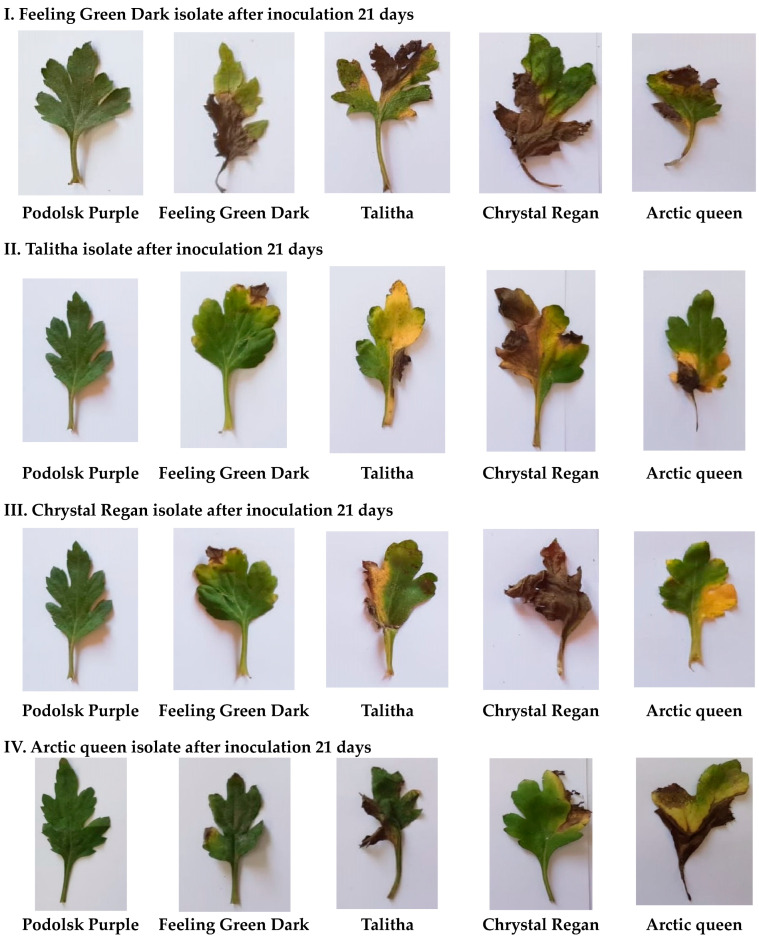
Responses of the different chrysanthemum cultivars shown as leaf symptoms 21 days after inoculation of the respective *A. alternata* isolates.

**Figure 5 plants-13-00252-f005:**
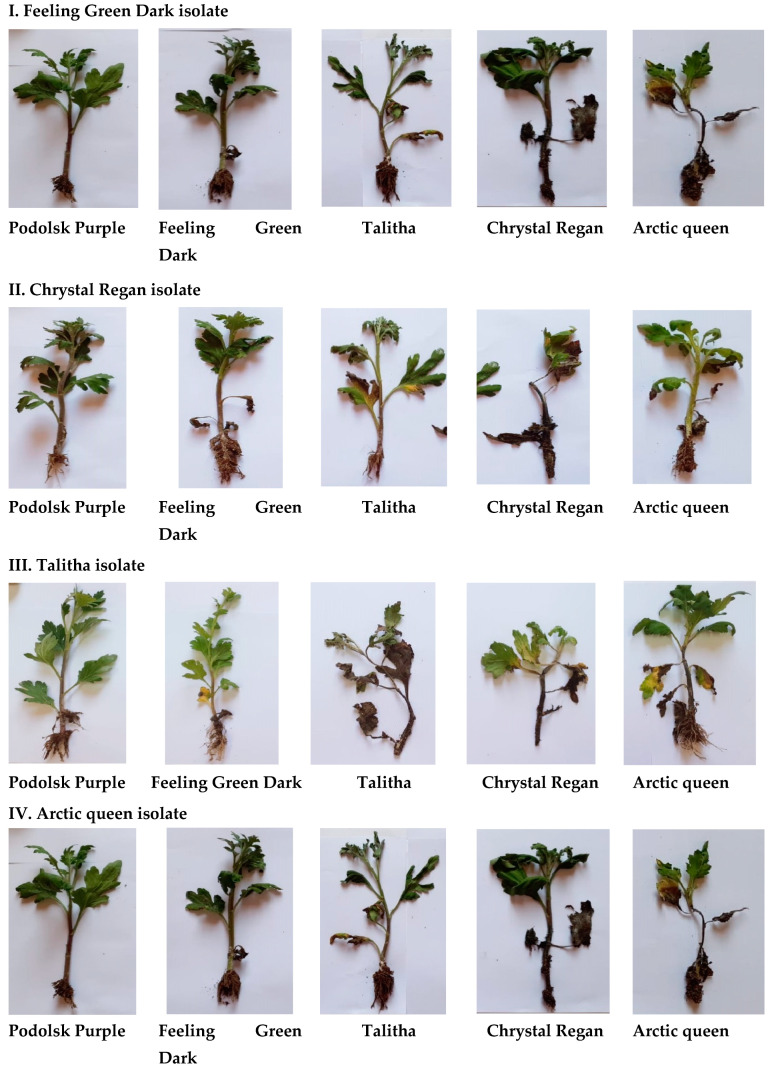
Response of the different chrysanthemum cultivars shown as symptoms on seedlings 21 days after inoculation of the respective *A. alternata* isolates.

**Figure 6 plants-13-00252-f006:**
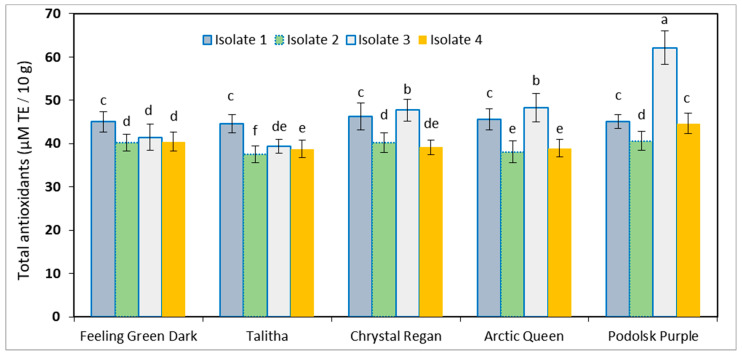
Antioxidant capacity of five chrysanthemum cultivars that were exposed to four isolates of *Alternaria alternata*. Means followed by the same letter in same column are not significantly different at the 0.05 level.

**Figure 7 plants-13-00252-f007:**
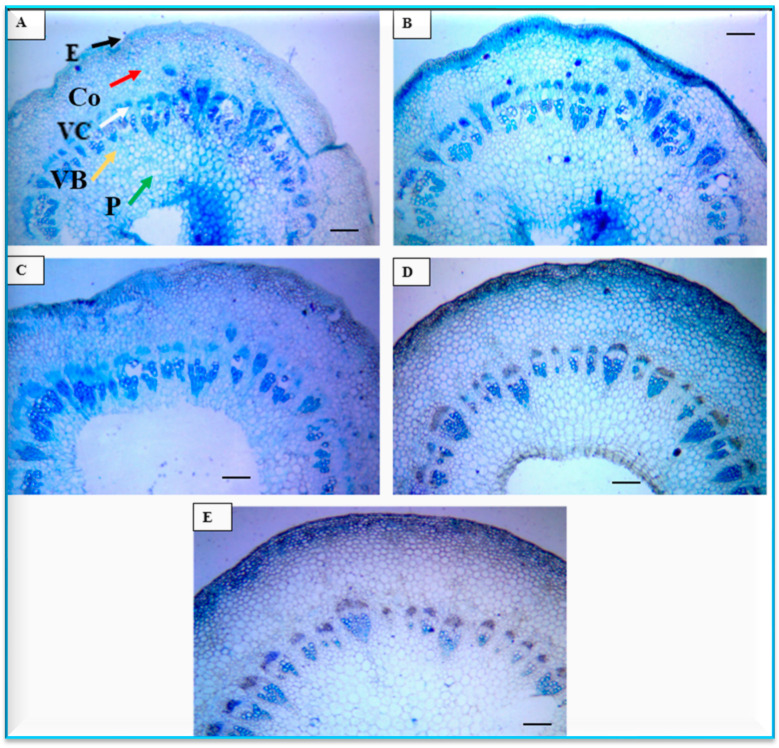
Histological micrograph of the studied chrysanthemum cultivar stems, for cultivars (**A**) Feeling Green Dark, (**B**) Talitha, (**C**) Chrystal Regan, (**D**) Arctic queen, and (**E**) Podolsk Purple. bars = 100 μm. Epidermis (E), Cortex (C), Vascular cylinder (VC), Vascular bundle (VB), Pith (P).

**Figure 8 plants-13-00252-f008:**
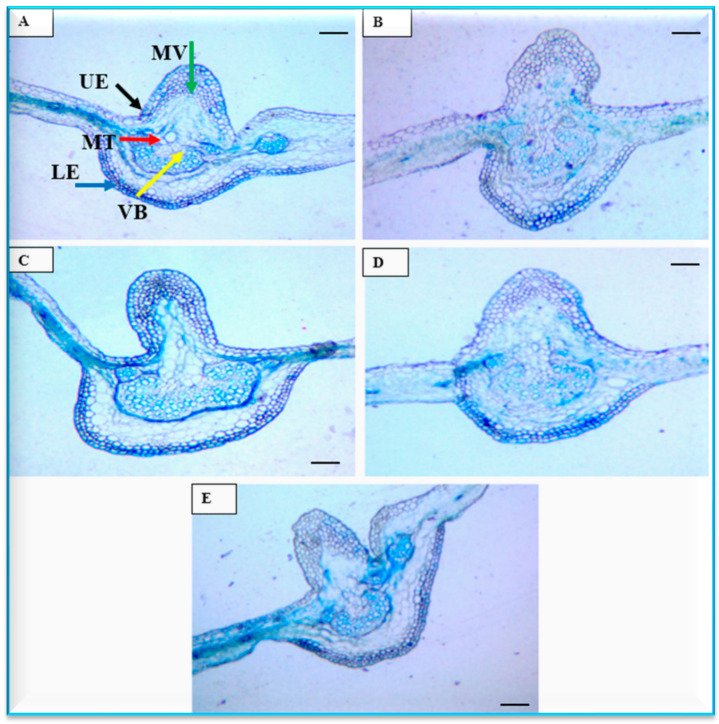
Histological micrograph of the studied chrysanthemum cultivar leaves, for cultivars (**A**) Feeling Green Dark, (**B**) Talitha, (**C**) Chrystal Regan, (**D**) Arctic queen, and (**E**) Podolsk Purple. bars = 100 μm. Upper epidermis (UE), Mesophyll tissue (MT), Vascular bundle (VB), Middle vein (MV), Lower epidermis (LE).

**Figure 9 plants-13-00252-f009:**
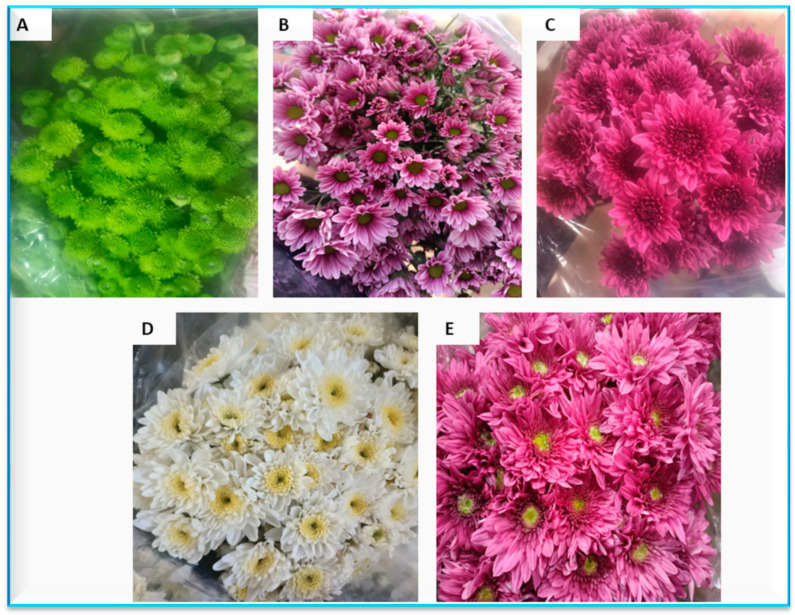
Chrysanthemum cultivars used in the experiments. (**A**) Feeling Green Dark, (**B**) Talitha, (**C**) Chrystal Regan, (**D**) Arctic Queen, and (**E**) Podolsk Purple.

**Table 1 plants-13-00252-t001:** Percent disease index values (PDI %) of the examined chrysanthemum cultivars inoculated with different isolates of Alternaria blight spot during the rooting stage.

Chrysanthemum Cultivars	Percent Disease Index (PDI %) Days After Inoculation			
	7	14	21	Mean
Inoculation with *A. alternata* isolate from cv **Feeling Green Dark**				
1- Feeling Green Dark	50.4 a ± 4.18	60.8 a ± 4.75	70.8 a ± 5.06	60.7 a ± 4.68
2- Talitha	00.0 d ± 0.00	20.4 c ± 1.93	40.5 c ± 3.12	20.3 d ± 2.04
3- Chrystal Regan	40.5 b ± 2.89	60.7 a ± 4.09	60.9 b ± 4.53	54.0 b ± 3.05
4- Arctic Queen	30.2 c ± 2.74	40.6 b ± 3.06	40.3 c ± 2.94	37.3 c ± 3.00
5- Podolsk Purple	00.0 d ± 0.00	00.0 d ± 0.00	00.0 d ± 0.00	00.0 e ± 0.00
F-test	**	**	**	**
Inoculation with *A. alternata* isolate from cv **Talitha**				
1- Feeling Green Dark	00.0 c ± 0.00	00.0 d ± 0.00	6.1 d ± 0.48	2.0 c ± 0.29
2- Talitha	20.3 b ± 1.76	48.6 a ± 4.33	84.7 a ± 6.03	51.2 a ± 4.84
3- Chrystal Regan	30.8 a ± 3.36	40.1 b ± 3.88	70.5 b ± 4.88	47.1 a ± 4.41
4- Arctic Queen	30.2 a ± 3.44	30.5 c ± 3.52	40.3 c ± 4.20	33.7 b ± 4.03
5- Podolsk Purple	00.0 c ± 0.00	00.0 d ± 0.00	00.0 e ± 0.00	00.0 c ± 0.00
F-test	**	**	**	**
Inoculation with *A. alternata* isolate from cv **Chrystal Regan**				
1- Feeling Green Dark	10.1 c ± 0.85	12.3 c ± 1.10	12.5 c ± 1.11	11.6 c ± 1.06
2- Talitha	8.5 c ± 0.69	10.2 c ± 0.88	14.7 c ± 1.13	11.1 c ± 1.12
3- Chrystal Regan	72.8 a ± 5.12	80.4 a ± 5.77	92.2 a ± 5.82	81.8 a ± 5.52
4- Art Queen	20.4 b ± 2.19	30.7 b ± 3.52	30.4 b ± 3.29	27.2 b ± 2.99
5- Podolsk Purple	00.0 d ± 0.00	00.0 d ± 0.00	00.0 d ± 0.00	00.0 d ± 0.00
F-test	**	**	**	**
Inoculation with *A. alternata* isolate from cv **Arctic queen**				
1- Feeling Green Dark	10.3 c ± 0.79	12.4 d ± 1.01	15.2 d ± 1.11	12.6 d ± 1.21
2- Talitha	12.2 c ± 0.94	20.3 c ± 2.17	26.4 c ± 2.29	20.0 c ± 2.22
3- Chrystal Regan	20.7 b ± 2.88	32.9 b ± 3.27	40.5 b ± 4.03	31.0 b ± 2.98
4- Arctic Queen	68.3 a ± 5.01	72.1 a ± 4.99	80.5 a ± 5.29	73.6 a ± 5.13
5- Podolsk Purple	00.0 d ± 0.00	00.0 e ± 0.00	00.0 e ± 0.00	00.0 e ± 0.00
F-test	**	**	**	**

** indicates significant differences at *p* values < 0.01, respectively, according to F. test. Means followed by the same letter in same column are not significantly different at “*p* = 0.05 level”, according to Duncan’s multiple range test. (±SD = standard deviation).

**Table 2 plants-13-00252-t002:** Effect of different isolates of *Alternaria alternata* on vegetative growth traits of Chrysanthemum seedlings.

Treatments		Number of Roots per Seedling	Number of Leaves per Seedling	Root Length (cm)	Root F.W. (g/seedling)	Seedling F.W. (g/seedling)
Isolates	Cultivars					
Isolate 1	Feeling Green Dark	26.04 bc ± 3.05	9.35 c ± 1.52	3.93 ab ± 0.85	0.31 b ± 0.05	2.12 b ± 0.48
	Talitha	25.27 c ± 1.00	8.71 d ± 0.58	3.56 b ± 0.47	0.30 b ± 0.03	2.10 b ± 0.24
	Chrystal Regan	23.50 d ± 3.79	7.73 e ± 1.53	2,60 d ± 0.05	0.27 bc ± 0.07	1.92 c ± 0.33
	Arctic Queen	24.50 d ± 1.25	8.77 d ± 0.58	3.16 c ± 0.76	0.29 b ± 0.09	2.14 b ± 0.24
	Podolsk Purple	30.10 ab ± 3.77	10.35 b ± 1.88	3.99 ab ± 0.45	0.35 a ± 0.07	2.42 ab ± 0.17
Isolate 2	Feeling Green Dark	28.11 b ± 6.24	10.12 b ± 0.58	3.79 b ± 1.00	0.23 cd ± 0.01	1.96 c ± 0.41
	Talitha	27.34 b ± 6.12	9.47 c ± 0.58	3.73 b ± 0.49	0.23 cd ± 0.04	1.94 c ± 0.11
	Chrystal Regan	25.57 c ± 6.24	8.49 d ± 1.52	2.77 c ± 0.50	0.20 d ± 0.08	1.76 d ± 0.40
	Arctic Queen	26.57 bc ± 5.03	9.54 c ± 3.21	3.33 b ± 0.50	0.22 cd ± 0.07	1.98 c ± 0.49
	Podolsk Purple	32.17 a ± 2.08	11.12 a ± 1.15	4.16 ab ± 0.06	0.27 bc ± 0.03	2.26 b ± 0.30
Isolate 3	Feeling Green Dark	21.95 e ± 3.77	9.22 c ± 1.73	3.93 ab ± 0.50	0.21 cd ± 0.01	1.69 de ± 0.19
	Talitha	21.17 e ± 4.51	8.57 d ± 0.58	3.81 b ± 0.50	0.21 cd ± 0.20	1.67 de ± 0.19
	Chrystal Regan	19.40 g ± 3.61	7.59 e ± 1.52	2.85 c ± 0.58	0.18 e ± 0.02	1.49 e ± 0.15
	Arctic Queen	20.40 f ± 1.53	8.62 d ± 0.58	3.40 b ± 0.92	0.20 d ± 0.05	1.71 d ± 0.39
	Podolsk Purple	26.00 bc ± 4.93	10.22 b ± 0.58	4.23 a ± 0.36	0.25 c ± 0.02	1.99 c ± 0.26
Isolate 4	Feeling Green Dark	22.75 de ± 1.54	9.35 c ± 0.58	4.06 ab ± 1.04	0.31 b ± 0.05	2.35 ab ± 0.27
	Talitha	21.97 e ± 4.93	8.71 d ± 1.52	3.93 ab ± 1.00	0.30 b ± 0.07	2.33 ab ± 0.21
	Chrystal Regan	20.20 f ± 3.51	7.73 e ± 2.00	2.97 c ± 1.00	0.27 bc ± 0.07	2.15 b ± 0.21
	Arctic Queen	21.20 e ± 1.53	8.77 d ± 1.00	3.53 b ± 0.32	0.29 b ± 0.08	2.36 ab ± 0.33
	Podolsk Purple	26.80 bc ± 5.27	10.35 b ± 1.73	4.36 a ± 0.92	0.35 a ± 0.04	2.65 a ± 0.43
Significance						
I (Isolates)		**	*	*	**	**
C (Cultivar)		**	**	**	**	**
I × C		**	*	*	*	**

Isolates are studied isolates from Feeling Green Dark, Talitha, Chrystal Regan, and Arctic queen; FW: fresh weight. * and ** indicate significant differences at *p* values < 0.01 and <0.05, respectively. Means followed by the same letter in the same column are not significantly different at “*p* = 0.05 level”, according to Duncan’s multiple range test. (±SD = standard deviation).

**Table 3 plants-13-00252-t003:** Effect of different isolates of *Alternaria alternata* on dry weight of roots and seedlings, chlorophyll fluorescence, and total chlorophyll content of Chrysanthemum seedlings.

Treatments		Root Dry Weight (g/seedling)	Seedling Dry Weight (g/seedling)	Total Chlorophyll Content (SPAD)	Fv/Fm
Isolates	Cultivars				
Isolate 1	Feeling Green Dark	0.10 a ± 0.01	0.42 b ± 0.04	18.70 e ± 1.24	0.79 a ± 0.06
	Talitha	0.09 a ± 0.01	0.41 b ± 0.04	18.62 e ± 2.17	0.79 a ± 0.06
	Chrystal Regan	0.08 a ± 0.01	0.38 c ± 0.05	17.58 f ± 1.91	0.79 a ± 0.00
	Arctic Queen	0.09 a ± 0.01	0.42 b ± 0.05	18.67 e ± 1.70	0.80 a ± 0.05
	Podolsk Purple	0.11 a ± 0.02	0.45 ab ± 0.04	20.09 c ± 1.90	0.80 a ± 0.07
Isolate 2	Feeling Green Dark	0.08 a ± 0.01	0.38 c ± 0.03	20.84 bc ± 2.07	0.79 a ± 0.06
	Talitha	0.07 a ± 0.01	0.37 c ± 0.03	20.77 bc ± 1.80	0.78 a ± 0.02
	Chrystal Regan	0.06 a ± 0.01	0.34 de ± 0.03	19.72 d ± 2.06	0.78 a ± 0.02
	Arctic Queen	0.07 a ± 0.01	0.38 c ± 0.04	20.81 bc ± 1.91	0.79 a ± 0.02
	Podolsk Purple	0.09 a ± 0.01	0.41 b ± 0.05	22.24 a ± 3.08	0.80 a ± 0.02
Isolate 3	Feeling Green Dark	0.07 a ± 0.01	0.33 de ± 0.02	19.39 d ± 5.05	0.79 a ± 0.02
	Talitha	0.06 a ± 0.01	0.32 e ± 0.03	19.31 d ± 3.16	0.78 a ± 0.01
	Chrystal Regan	0.05 a ± 0.01	0.28 f ± 0.03	18.27 ef ± 0.71	0.78 a ± 0.01
	Arctic Queen	0.06 a ± 0.01	0.32 de ± 0.05	19.36 d ± 1.17	0.79 a ± 0.02
	Podolsk Purple	0.08 a ± 0.01	0.35 d ± 0.06	20.78 bc ± 3.04	0.80 a ± 0.02
Isolate 4	Feeling Green Dark	0.09 a ± 0.02	0.44 ab ± 0.06	21.60 b ± 4.01	0.79 a ± 0.01
	Talitha	0.08 a ± 0.01	0.43 ab ± 0.05	21.52 b ± 3.36	0.78 a ± 0.01
	Chrystal Regan	0.08 a ± 0.01	0.39 bc ± 0.07	20.48 c ± 4.01	0.78 a ± 0.01
	Arctic Queen	0.08 a ± 0.01	0.43 ab ± 0.04	21.57 b ± 3.38	0.79 a ± 0.02
	Podolsk Purple	0.10 a ± 0.01	0.46 a ± 0.06	22.99 a ± 4.11	0.80 a ± 0.02
Significance					
I (Isolates)		**	**	**	*
C (Cultivar)		*	*	*	**
I × C		NS	**	*	NS

Isolates are studied isolates from Feeling Green Dark, Talitha, Chrystal Regan, and Arctic queen, Fv/Fm = photochemical efficiency of PSII, where maximal variable fluorescence (Fv = Fm − F0), F0= minimal fluorescence. NS, * and ** indicate non-significant and significant differences at *p* values < 0.01 and <0.05, respectively. Means followed by the same letter in same column are not significantly different at “*p* = 0.05 level”, according to Duncan’s multiple range test. (±SD = standard deviation).

**Table 4 plants-13-00252-t004:** Anatomical characters of the stems of studied cultivars under exposition to *Alternaria alternata*.

Cultivars	Thickness of Cuticle (µm)	Thickness of Epidermis (µm)	Cortex Thickness (µm)	Vascular Cylinder Diameter (µm)	Stem Diameter (µm)	Diameter of Xylem Vessel (µm)
Feeling Green Dark	1.67 b ± 0.15	4.90 a ± 0.67	52.96 c ± 5.55	248.60 c ± 9.00	365.98 d ± 4.80	13.42 b ± 0.82
Talitha	2.51 b ± 0.35	4.57 a ± 0.79	52.52 c ± 2.08	287.24 a ± 0.96	403.85 c ± 3.89	14.16 b ± 0.87
Chrystal Regan	2.58 b ± 0.38	4.21 a ± 0.53	75.90 b ± 1.95	278.21 a ± 0.98	428.04 b ± 6.51	13.26 b ± 0.61
Arctic Queen	2.55 b ± 0.81	4.04 a ± 1.25	70.39 b ± 2.45	263.26 b ± 2.62	408.58 c ± 4.86	12.95 c ± 1.83
Podolsk Purple	3.94 a ± 0.69	3.97 a ± 0.91	99.03 a ± 4.48	279.06 a ± 8.05	490.760 a ± 2.06	17.09 a ± 0.79
F. test	**	NS	**	**	**	**

NS, ** indicate non-significant and significant differences at *p* values < 0.05. Means followed by the same letter in the same column are not significantly different at “*p* = 0.05 level”, according to Duncan’s multiple range test. (±SD = standard deviation).

**Table 5 plants-13-00252-t005:** Anatomical characters of leaves of studied cultivars under exposition of *Alternaria alternata*.

Cultivars	Thickness of Cuticle (µm)	Thickness of Epidermis (µm)	Thickness of Mesophyll Tissue (µm)	Thickness of Lamina (µm)	Thickness of Middle Vein (µm)	Diameter of Vascular Bundle (µm)
Feeling Green Dark	3.08 c ± 0.60	9.64 bc ± 1.26	51.12 a ± 2.35	65.23 a ± 5.97	150.80 d ± 0.49	69.15 b ± 1.91
Talitha	6.13 a ± 0.69	10.96 b ± 1.40	27.41 b ± 3.57	46.98 b ± 5.70	208.91 a ± 0.46	109.78 a ± 15.07
Chrystal Regan	5.42 ab ± 0.35	7.63 c ± 1.03	17.81 c ± 2.61	23.84 c ± 3.25	209.29 a ± 2.02	118.20 a ± 13.62
Arctic Queen	5.76 a ± 0.95	8.43 bc ± 2.06	28.93 b ± 3.98	39.69 b ± 4.39	197.44 b ± 2.93	109.89 a ± 1.75
Podolsk Purple	4.33 b ± 0.51	14.26 a ± 1.07	45.20 a ± 4.70	68.53 a ± 1.77	171.76 c ± 2.08	81.90 b ± 2.92
F. test	**	**	**	**	**	**

** indicate significant differences at *p* values <0.05. Means followed by the same letter in same column are not significantly different at “*p* = 0.05 level”, according to Duncan’s multiple range test. (±SD = standard deviation).

## Data Availability

Data are contained within the article.

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
