# Peer review of "Evaluation of Five Chrysanthemum morifolium Cultivars against Leaf Blight Disease Caused by Alternaria alternata at Rooting and Seedling Growth Stages"

_plants, 2024, doi:10.3390/plants13020252_

Round 1

Reviewer 1 Report

Comments and Suggestions for Authors

Dear Authors

 I am writing to you in regard to manuscript entitled: “Evaluation of Five Chrysanthemum morifolium Cultivars Against Leaf Blight Disease Caused by Alternaria alternata at Rooting and Seedling Growth Stages”

I think that this manuscript is sufficiently conceived and organized in every section opening to a clear scenario of actual topic. Given the increasing crop importance of chrysanthemum in Egypt and worlwide, I also think the paper’s topic is very interesting and it fully accomplishes the scope of Plants.

The authors claimed a significant variability in cultivar susceptibility to early blight caused by A. alternata on Chrysanthemum

Nevertheless, the quality of the paper is not still enough for a direct publication. Comprehensively, the paper include new and interesting information that could be useful for the researchers and technicians and it should be published following deep and major revisions:

- Statistical analyses section should be furtherly implemented. Many details and information are missing. For example, standard deviation (SD) or Standard Error values should be supplemented in all tables after the means. Moreover, since the experiment has been repeated, is there any parameter or analysis that shows the data of the two trials are similar or comparable?

- I strongly suggest also a deep English revision made by a native speaker or by a colleague fluent in English. Indeed, the meanings of many sentences, are too long or hard to follow for the reader.

- many Figures (E.g. Figure 1A,B,C, E) are not clear because they are at very low resolution. Please replace or improve these pictures.

- Reference section should be checked accordingly to journal guidelines. Moreover, some references should be added.

Many modifications and comments should be performed or addressed prior to publication in the annotated PDF.

Please consider in attachment the annotated PDF and fulfill point-by-point all included requirements.

 Yours sincerely

Comments on the Quality of English Language

I strongly suggest also a deep English revision made by a native speaker or by a colleague fluent in English. Indeed, the meanings of many sentences, are too long or hard to follow for the reader.

Author Response

Dear Reviewer 1#

We thank you so much for your time and revising our MS. We totally respect your opinion, all your comments, and encouragements, thanks a lot again.

Comments and Suggestions for Authors

Dear Authors

I am writing to you in regard to manuscript entitled: “Evaluation of Five Chrysanthemum morifolium Cultivars Against Leaf Blight Disease Caused by Alternaria alternata at Rooting and Seedling Growth Stages”

Response to the Reviewer:

Again, many thanks for your so positive feedback.

I think that this manuscript is sufficiently conceived and organized in every section opening to a clear scenario of actual topic. Given the increasing crop importance of chrysanthemum in Egypt and worldwide, I also think the paper’s topic is very interesting and it fully accomplishes the scope of Plants.

Response to the Reviewer:

Again, many thanks for your so positive feedback.

The authors claimed a significant variability in cultivar susceptibility to early blight caused by A. alternata on Chrysanthemum

Nevertheless, the quality of the paper is not still enough for a direct publication. Comprehensively, the paper includes new and interesting information that could be useful for the researchers and technicians and it should be published following deep and major revisions:

Response to the Reviewer:

Again, many thanks for your so positive feedback.

- Statistical analyses section should be furtherly implemented. Many details and information are missing. For example, standard deviation (SD) or Standard Error values should be supplemented in all tables after the means. Moreover, since the experiment has been repeated, is there any parameter or analysis that shows the data of the two trials are similar or comparable?

Response to the Reviewer:

Again, many thanks for your so positive feedback. The standard deviation (SD) or Standard Error values were added to all Tables in the revised MS. Thanks a lot.

- I strongly suggest also a deep English revision made by a native speaker or by a colleague fluent in English. Indeed, the meanings of many sentences, are too long or hard to follow for the reader.

Response to the Reviewer:

Again, many thanks for your so positive feedback. The revised MS was edited and the edited file can be found in the supplementary files to show the editing of the MS using the tracking system. Thanks a lot.

- many Figures (E.g. Figure 1A,B,C, E) are not clear because they are at very low resolution. Please replace or improve these pictures.

Response to the Reviewer:

Again, many thanks for your so positive feedback. The figures were replaced with higher resolution and we will send the ppt file to the journal after accepting our MS to present the best version of these figures. Thanks again.

- Reference section should be checked accordingly to journal guidelines. Moreover, some references should be added.

Response to the Reviewer:

Again, many thanks for your so positive feedback. The list of refs. was double checked and some refs. besides your suggested refs. were added as the reviewer asked, thanks a lot.

Many modifications and comments should be performed or addressed prior to publication in the annotated PDF.

Please consider in attachment the annotated PDF and fulfill point-by-point all included requirements.

Yours sincerely

Response to the Reviewer:

Again, many thanks for your so positive feedback. Your following comments were responded one by one in the following table:

The comment of reviewer

Response to the reviewer

Line 32: please rephrase "are in accordance with this experiment"

Was changed after editing, thanks

Line 40: please rephrase with "findings"

Replaced, thanks.

Lines 51 and 52:

The sentence was revised, thanks

Lines 65-67:

These information are incomplete. Among key fungal diseases of Chrysanthemum morifolium I suggested to the authors to include in this brief description at the least rust (by Puccinia horiana) and the Verticillium wilt (by Verticillium dahliae) with relative recent references, i.e. O'Keefe and Davis 2015 [14] and Castello et al. 2022 [15]

Following the references in extenso to add:

14. O'Keefe, G.; Davis, D. D. Morphology of Puccinia horiana, Causal Agent of Chrysanthemum White Rust, Sampled From Naturally Infected Plants. Plant Dis. 2015 99(12), 1738–1743.

15. Castello, I.; D’Emilio, A.; Baglieri, A.; Polizzi, G.; Vitale, A. Management of Chrysanthemum Verticillium Wilt through VIF Soil Mulching Combined with Fumigation at Label and Reduced Rates. Agriculture 2022, 12, 141

The suggested refs. were added and a part also was added to the revised MS, Thanks.

Added refs. are:

O'Keefe, G.; Davis, D. D. Morphology of Puccinia horiana, Causal Agent of Chrysanthemum White Rust, Sampled from Naturally Infected Plants. Plant Dis. 2015, 99(12), 1738–1743.

Castello, I.; D’Emilio, A.; Baglieri, A.; Polizzi, G.; Vitale, A. Management of Chrysanthemum Verticillium Wilt through VIF Soil Mulching Combined with Fumigation at Label and Reduced Rates. Agriculture, 2022, 12, 141

Gao, G.; Jin, R.; Liu, D.; Zhang, X.; Sun, X.; Zhu, P.; Mao, H. CmWRKY15-1 Promotes Resistance to Chrysanthemum White Rust by Regulating CmNPR1 Expression. Front. Plant Sci, 2022, 13:865607. doi: 10.3389/fpls.2022.865607

Sumitomo, K.; Shirasawa, K.; Isobe, S.N.; Hirakawa, H.; Harata, A.; Kawabe, M.; Yagi, M.; Osaka, M.; Kunihisa, M.; Taniguchi, F. DNA marker for resistance to Puccinia horiana in chrysanthemum (Chrysanthemum morifolium Ramat.) "Southern Pegasus". Breed Sci, 2021, 71(2):261-267. doi: 10.1270/jsbbs.20063.

Bi, M.; Li, X; Yan, X.; et al. Chrysanthemum WRKY15-1 promotes resistance to Puccinia horiana Henn. via the salicylic acid signaling pathway. Hortic Res, 2021, 8, 6. https://doi.org/10.1038/s41438-020-00436-4

Zhang S, Miao W, Liu Y, Jiang J, Chen S, Chen F, Guan Z. Jasmonate signaling drives defense responses against Alternaria alternata in chrysanthemum. BMC Genomics. 2023, 24(1):553. doi: 10.1186/s12864-023-09671-0.

Munilakshmi, R.; Reddy, B.A.; Hubballi, M. et al. Characterization of Puccinia horiana causing Chrysanthemum rust disease and its management by altering planting date and foliar application of fungicide. Indian Phytopathol, 2023, 76, 437–445. https://doi.org/10.1007/s42360-022-00574-w

Chen, Q.; Kuang, A.; Wu, H.; Liu, D.; Zhang, X.; Mao, H. Physiological response of CmWRKY15-1 to chrysanthemum white rust based on TRV-VIGS. Front Plant Sci, 2023, 14:1140596. doi: 10.3389/fpls.2023.1140596.

Line 96:

do you mean "crop season"?

Corrected according to your suggestion, thanks.

Lines 102-106:

Too long sentence. Please divided it two short and clear little sentences.

Revised and corrected based your advice, thanks.

Lines 119-120:

this sentence is very confuse. What do you mean "diseased fungi". Perhaps fungi causal agent of early blight. Please clarify it.

Corrected in the revised Ms into:

The four isolates as diseased fungi were collected and isolated from diseased plants with observed symptoms with Alternaria leaf spot indicators.

Line 123:

sterilized

Changed into “sterilized” according to the advice of reviewer, thanks

Line 152:

"causal agent of disease"?

Changed into “causal agent of disease” according to the advice of reviewer, thanks

Figure 1:

except for Figure D I think that all photos are in low resolution. Please improve or replace these photos if it is possible.

All photos were replaced by ones in a high resolution, thanks.

Line 174:

for example a visual-score scale with different classes should be very useful for the reader

We added a figure including this visual score, thanks.

Line 181:

please replace with " sum of numerical rating scores"

Done, thanks.

Line 122-123:

please replace with "(isolate as first factor and cultivar as second factor)"

Corrected, and revised, thanks.

Line 125:

what do you mean" perhaps, averaged?

Corrected, and revised, thanks.

Line 126:

please rephrase with "a two-way ANOVA approach of the CoStat program...."

Corrected, and revised, thanks.

Line 234:

please write Fig. 2A, B

Corrected, and revised, thanks.

Line 258:

please replace with fungal colonies, and “four”

Corrected, and revised, thanks.

Line 261:

The average value should be followed by +/- SD (standard deviation) or SE or SEM (Standard Error of the Mean). Please provide it.

Provided in the revised MS, thanks

Table 1:

"inoculation with A. alternata isolate from cv Feeling green dark"

Corrected, and revised, thanks.

Table 2:

see previous comment. Please, add +/-SD or SE or SEM

Corrected, and revised, thanks.

References:

please check all references accordingly to guidelines of journal. for example abbreviations of journals with or without fullstop. Please provide it.

Corrected, and revised, thanks.

Comments on the Quality of English Language

I strongly suggest also a deep English revision made by a native speaker or by a colleague fluent in English. Indeed, the meanings of many sentences, are too long or hard to follow for the reader.

Response to the Reviewer:

Again, as mentioned above, many thanks for your so positive feedback. The revised MS was edited and the edited file can be found in the supplementary files to show the editing of the MS using the tracking system. Thanks a lot.

Reviewer 2 Report

Comments and Suggestions for Authors

The presented article is of great practical importance. Climate changes in recent decades have not made it easier to grow plants, disrupting the production technologies developed so far and increasing pathogenic factors. The article is an excellent response to currently emerging problems. The analysis of vegetative, physiological and histological parameters as well as antioxidant capacity allowed us to fully answer the question of what factors are responsible for Chymanthemum's resistance to A. alternata.

I believe that in the methodology chapter it would be worth adding a description of the chrysanthemum varieties assessed in this study. Are these taxa commonly cultivated in Egypt? What percentage do they constitute in the cultivation structure of this species?

Comments on the Quality of English Language

I have no objections.

Author Response

Reviewer 2#

Dear Reviewer 2#

We thank you so much for your time and revising our MS. We totally respect your opinion, and encouragements, thanks a lot for your so kind words.

Comments and Suggestions for Authors

The presented article is of great practical importance. Climate changes in recent decades have not made it easier to grow plants, disrupting the production technologies developed so far and increasing pathogenic factors. The article is an excellent response to currently emerging problems. The analysis of vegetative, physiological and histological parameters as well as antioxidant capacity allowed us to fully answer the question of what factors are responsible for Chymanthemum's resistance to A. alternata.

Response to the Reviewer:

Again, many thanks for your so positive feedback.

I believe that in the methodology chapter it would be worth adding a description of the chrysanthemum varieties assessed in this study. Are these taxa commonly cultivated in Egypt? What percentage do they constitute in the cultivation structure of this species?

Response to the Reviewer:

We added this part based on your advice, thanks.

2.2 Description of the chrysanthemum varieties

Are these chrysanthemum varieties commonly cultivated in Egypt? What percentage do they constitute in the cultivation structure of this species? These cultivars are grown throughout the year in Egypt to produce flowers, but some cultivars, such as cv Talitha is preferred to be planted in the winter during October, whereas cv Arctic queen is preferable in the summer during April. The cultivars used in this study differ in terms of plant height, stem diameter, and flower color. Concerning cv Feeling green dark, it has a high stem, strong stem with large thickness and green flowers, whereas cv Talitha is characterized by a medium-height stem with thin thickness and single, light purple flowers. Regrading, cvs Chrystal Regan, Arctic queen and Podolsk Purple are characterized by strong stems with a large thickness and their flowers are in light red colors, white flowers and purple, respectively. The percentage of cultivation of studied cultivars varies within different farms according to the planting season and the marketing contract, but in general the average for 10 farms in Menoufia and Gharbia governorates, the cultivation rates of Feeling green dark, Talitha, Chrystal Regan, Arctic queen and Podolsk Purple cultivars are 3-6%, 3-5%, 8-10%, 20-25% and 10-15%, respectively.

Comments on the Quality of English Language

I have no objections.

Response to the Reviewer:

Many thanks again for your so positive feedback.

Round 2

Reviewer 1 Report

Comments and Suggestions for Authors

The Authors have properly addressed my suggestions.

Author Response

Dear Reviewer

many thanks for your time and comments

really your comments already improved our MS

thanks again